# A Change of Heart:
# Improving Speech Emotion Recognition through Speech-to-Text Modality Conversion

**Zeinab Sadat Taghavi, Ali Satvati & Hossein Sameti**
Department of Computer Engineering
Sharif University of Technology
Tehran, Iran
`{zeinabtaghavi,alisatvati,sameti}@sharif.edu`

## Abstract

Speech Emotion Recognition (SER) is a challenging task. In this paper, we introduce a modality conversion concept aimed at enhancing emotion recognition performance on the MELD dataset. We assess our approach through two experiments: first, a method named Modality-Conversion that employs automatic speech recognition (ASR) systems, followed by a text classifier; second, we assume perfect ASR output and investigate the impact of modality conversion on SER, this method is called Modality-Conversion++. Our findings indicate that the first method yields substantial results, while the second method outperforms state-of-the-art (SOTA) speech-based approaches in terms of SER weighted-F1 (WF1) score on the MELD dataset. This research highlights the potential of modality conversion for tasks that can be conducted in alternative modalities.

## 1 Introduction

Emotion recognition from speech is challenging due to audio complexity and expression variability. SOTA baseline approaches rely on a one-modality approach (4). However, if there is one modality that performs better than speech (10), can converting the modality improve results when using the modality that had lower performance on the same dataset? In this paper, we examine this idea with two experiments: first, we introduce a model named Modality-Conversion that employs ASR and then uses a text classifier; second, we assume a perfect ASR output and examine the impact of modality conversion on emotion recognition; we call this method Modality-Conversion++. While the first method achieved significant results compared to SOTA, the second method improves emotion recognition performance even further, compared to SOTA speech-based approaches on the MELD dataset from the TV series *Friends* (4). The study demonstrates the potential of modality conversion and text-based techniques for enhancing emotion recognition from speech.[1]

## 2 Related Works

Emotion recognition from speech has been studied extensively, with many approaches focusing on acoustic features extracted from speech signals (3) (5) (6) (4) (10). In the context of emotion recognition, researchers have also explored the use of text-based features (13) (8). Multi-modality approaches, which combine both speech and text-based features, have also been proposed and have shown promising results in emotion recognition and show that on the same dataset, text-based approaches achieve better results than speech-based approaches (7) (10) (12) (16) (11) (14) (2) (17) (15). However, to the best of our knowledge, no one has explored using modality conversion to improve emotion recognition from any of the modalities on the MELD dataset.

---

[1] The implementations are available at: `https://github.com/ICLR2023AChangeOfHeart/meld-modality-conversion`

## 3 METHOD

In this study, we utilized the MELD dataset consisting of audio-visual clips. In the first method, we extracted the audio from the clips and then used the Vosk API [2] for ASR to convert the audio clips to text (9). In the second method, we assumed an ideal ASR with optimal accuracy and used the gold transcript of each speech track. For both methods, we used a RoBERTa-base text classification model, fine-tuned on the modality converted text transcripts with emotion labels (1). We evaluated the performance of our proposed modality conversion approach, as shown in Figure 1, using the WF1 metric.

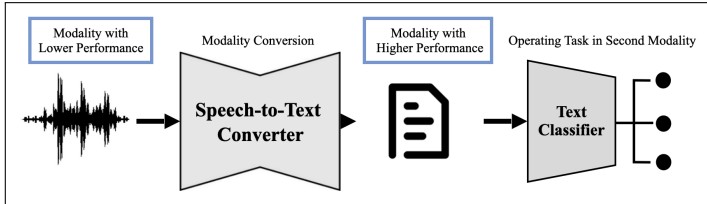

Figure 1: Our proposed idea about the emotion recognition pipeline consists of two main steps: speech-to-text conversion (Vosk in the first method, or an ideal converter in the second method), text classification in the second modality (fine-tuned RoBERTa model).

## 4 RESULTS

We compared the performance of our proposed modality conversion approach with the current SOTA speech-based models on the MELD dataset. The results, as presented in Table 1, show that our first method achieved a competitive WF1 score of 43.1%, which is even higher than SpeechFormer (4). Moreover, our second method outperforms the baseline models, achieving a WF1 score of 60.4%. These results demonstrate the effectiveness of our proposed modality conversion approach for emotion recognition from speech.

| Method | Input Modality | Using Modality Conversion | WF1(%) |
|---|---|---|---|
| SpeechFormer (7) | Speech | - | 41.9 |
| SpeechFormer++ (5) | Speech | - | 47.0 |
| DWFormer (3) | Speech | - | 48.5 |
| **DST (6)** | **Speech** | **-** | **48.8** |
| Modality-Conversion | Speech | Converting to Text Modality | 43.1 |
| **Modality-Conversion++** | **Speech** | **Converting to Text Modality** | **60.4** |

Table 1: Comparison of different methods for SER.

## 5 CONCLUSION

Based on our experiments on the MELD dataset, we proposed an alternative to traditional approaches: a modality conversion idea for tasks that have better performance on one modality than others, especially for SER. We examined this idea with two methods: first, the Vosk API for ASR as modality conversion, and second, considering an ideal modality conversion stage by using gold text. Finally, both methods employed a pre-trained RoBERTa language model for emotion recognition on the text transcripts. Our approach resulted in a WF1 score of 43.1% for the first method and 60.4% for the second method, which is an 11.6% improvement over the SOTA speech-based models on the same dataset, showing the potential of modality conversion idea.

---

[2]https://github.com/alphacep/vosk-api

ACKNOWLEDGEMENTS

We would like to acknowledge the support and resources provided by our institution for this research. We would also like to thank the creators of the MELD dataset, as well as the developers of the Vosk API and the pre-trained RoBERTa model. Lastly, we would like to express our gratitude to Soroush Gooran for his valuable insights and assistance throughout the course of this project.

URM STATEMENT

The authors acknowledge that at least one key author of this work meets the URM criteria of ICLR 2023 Tiny Papers Track.

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

APPENDIX

Our results demonstrate the potential of using modality conversion techniques and text—based features for enhancing emotion recognition from speech. Our approach offers an alternative to traditional speech—based approaches, which rely solely on analyzing audio signal features. Using text—based techniques, we can overcome some of the challenges posed by audio complexity and expression variability.

Future work can explore different ASR systems and NLP techniques for further improving the performance of the modality conversion approach. Additionally, incorporating visual cues and other modalities can be explored to enhance emotion recognition accuracy even further. Overall, our study highlights the promising direction of modality conversion and multi—modal approaches for improving emotion recognition from speech.

